# Policy Debates Regarding Nicotine Vaping Products in Australia: A Qualitative Analysis of Submissions to a Government Inquiry from Health and Medical Organisations

**DOI:** 10.3390/ijerph16224555

**Published:** 2019-11-18

**Authors:** Daniel A Erku, Kylie Morphett, Kathryn J Steadman, Coral E Gartner

**Affiliations:** 1School of Pharmacy, The University of Queensland, 20 Cornwall Street, Woolloongabba 4102, Queensland, Australia; k.steadman@uq.edu.au; 2School of Public Health, The University of Queensland, Herston Road, Herston 4006, Queensland, Australia; k.morphett@uq.edu.au (K.M.); c.gartner@uq.edu.au (C.E.G.); 3Queensland Alliance for Environmental Health Sciences, The University of Queensland, 20 Cornwall Street, Woolloongabba 4102, Queensland, Australia

**Keywords:** e-cigarettes, regulation, policy analysis, Australia

## Abstract

Australia has maintained a highly restrictive regulatory framework for nicotine vaping products (NVPs) and the regulatory approach differs from most other high income countries. This paper employed a thematic analysis to assess policy consultation submissions made to a government inquiry regarding use and marketing of NVPs. We included in the analysis submissions (*n* = 40) made by Australian institutions that influence or contribute to health policy-making including government agencies, health bodies and charities (*n* = 23), and public health academics and healthcare professionals (*n* = 18). Submissions from commercial entities and consumers were excluded. The majority of submissions from representatives of government agencies, health bodies and charities recommended maintaining current restrictions on NVPs. Arguments against widening access to NVPs included the demand for long-term evidence on safety and efficacy of an unusually high standard. There was widespread support for restrictions on sales, advertising and promotion, with most submissions supporting similar controls as for tobacco products. In contrast, the majority of individual submissions from healthcare professionals and public health academics advocated for widening access to NVPs for smokers and emphasized the potential benefits of smokers switching to vaping and the policy incoherence of regulating less harmful nicotine products more strictly than tobacco cigarettes. Progress in resolving the policy debate concerning NVP regulation in Australia will require policy makers, clinicians and the public health community to engage in a meaningful dialogue which gives due consideration to both intended and unintended consequences of proposed policies.

## 1. Introduction

Since their introduction into the market in the mid-2000s, the use of nicotine vaping products (NVPs) (also known as e-cigarettes or electronic nicotine delivery systems) has increased rapidly [1,2,3,4]. These products have now become more popular than nicotine replacement therapies (NRTs) for smoking cessation in some countries such as the UK [5]. Yet, there has been continued debate over the most appropriate regulatory framework for these products and whether their use should be encouraged for harm reduction purposes, with profoundly divergent standpoints evident in the public health field [6]. Government agencies in some countries such as the UK and New Zealand have not only endorsed a role for NVPs in reducing tobacco related harm, but are also proactively encouraging smokers to switch to vaping, including supporting stop smoking services to become ‘vaping friendly’ [7]. In the UK, a dual regulatory pathway exists for NVPs as either a medicine or a consumer product under the EU Tobacco Product Directive (TPD) [8]. Similarly, NVPs can be marketed as either medicines, if health claims are made, or tobacco products in New Zealand following the outcome of a court case in 2018.

Australia, on the other hand, has maintained a highly restrictive regulatory framework for NVPs. Most jurisdictions in Australia allow nicotine-free vaping products to be used and sold under similar conditions as imposed on combustible cigarettes [9]. Nicotine in non-therapeutic products is classified by the Therapeutic Goods Administration (TGA)’s Poisons Standard as a dangerous poison (Schedule 7). Nicotine in preparations for human therapeutic use are either an unscheduled medicine (can be sold in general retail outlets) when in oromucosal and transdermal delivery systems for smoking cessation, or a prescription only medicine (Schedule 4) for all other human therapeutic products Combustible tobacco products are exempt from scheduling [10]. As a result, the only way for Australians to access NVPs containing nicotine is for therapeutic purposes with a medical prescription (Schedule 4 medicine). However, since there are no NVPs registered in the Australian Register of Therapeutic Goods, NVPs can only be obtained through personal importation of up to three months’ supply from overseas, the Special Access Scheme, the Authorized Prescriber Scheme, or through extemporaneous compounding [11]. All of these pathways require the person to hold a valid prescription from a medical practitioner registered in Australia. There have been a number of government inquiries and unsuccessful attempts to liberalize the regulation of NVPs in Australia. These include a 2016 application from a consumer organization to the TGA to “exempt nicotine from Schedule 7 at concentrations of 3.6 per cent or less of nicotine for self-administration with an electronic nicotine delivery system for the purpose of tobacco harm reduction” [12], a Senate inquiry (‘Red Tape Committee’, October 2016) that examined the impact of government regulation on tobacco retailing including NVP marketing in Australia [13], and a Senate inquiry into the Vaporized Nicotine Products Bill (August 2017) [14], which was tabled by federal cross-bench senators. Most recently, the 2017 House of Representatives Inquiry Into the Use and Marketing of E-cigarettes and Personal Vaporizers in Australia [15] has been completed and the final report published.

For decades, the public health community across the globe have coalesced around a shared policy priority of ending the tobacco epidemic, and successfully advocated for adopting strong tobacco control policies. However, profoundly divergent standpoints are evident in the public health field when it comes to NVPs. A recent analysis of health policy positions in Scotland regarding NVP regulation found that although most policy actors agreed on some policy issues, such as age-of-sale restrictions, they were divided with regard to the harms and benefits of NVPs and on the most appropriate regulatory framework [16]. However, how the public health community in Australia have framed the NVP-related policy debate has not been systematically explored. In this study, we analyzed written submissions to and the transcripts of public hearings of the 2017 House of Representatives Inquiry from government health agencies, health charities, individual healthcare professionals (HCPs), public health academics, and organizations representing health and medical professionals (peak health bodies) with the following research questions: (1) What are the views of medical and health organizations, HCPs and public health academics regarding the safety and efficacy of NVPs? (2) What are the policy frameworks and preferred regulatory options that have been recommended by these organizations and individuals and what are the discourses used to describe them? (3) In what ways has the framing of scientific evidence about these products been used to justify the proposed regulations?

## 2. Materials and Methods

In 2017, the Standing Committee on Health, Aged Care and Sport invited the public and interested organizations to make submissions to an inquiry into the use and marketing of electronic cigarettes and personal vaporizers. The terms of reference included comment about:The use and marketing of e-cigarettes and personal vaporizers to assist smokers to quit;The health impacts of the use of e-cigarettes and personal vaporizers;International approaches to legislating and regulating e-cigarettes and personal vaporizers; andThe appropriate regulatory framework for Australia and any other related matters.

The Inquiry received 352 submissions from government and non-government organizations, academics, individual health care professionals, for profit businesses (including the tobacco industry), as well as members of the public. The Standing Committee held three public hearings, and three private briefings were also held with witnesses from the UK, the transcripts of which were later published. In this study, we analyzed written submissions from (1) institutions and organizations involved in health policy making in Australia (including government agencies, health charities and peak health bodies) and (2) individual university public health academics and HCPs based in Australia. If these organizations or individuals also participated in a public hearing, the relevant section of the transcript from the hearing was also analyzed (Table 1). Excluded were submissions from commercial entities (including tobacco companies), consumers and submissions from individuals or institutions based outside of Australia.

### Data Analysis

The first author (D.A.E) analyzed the data using thematic analysis, as per the procedures outlined by Braun and Clarke [17]. First, line by line reading of the submissions and interview transcripts was performed to establish familiarity with the data. Guided by our research aims, a set of pre-specified codes were then generated which were merged into a higher level ‘themes’ based on similarity, representing a broader topic. In addition, an inductive approach was employed to identify new and emergent themes as we coded the data. In order to ensure validity and transparency, a second author (K.M.) independently coded a subset of submissions and generated codes. As the coding and analysis progressed, the themes were compared and cross-examined between the two coders. Any disagreements between the coders were addressed through discussion and consensus. Coding was managed with NVivo 12 software (QSR International Pty Ltd, Melbourne, Australia).

## 3. Results

A total of 40 submissions were included in the analysis: nine from government bodies, seven from peak health bodies, seven from health charities, and 18 from individual HCPs and public health academics. These ranged from two to 16 pages in length. In addition, 19 of the submissions were supplemented with interview transcripts from the public hearing (Table 1).

An overwhelming majority of HCPs and public health academics who made individual submissions (83%) endorsed the role of NVPs as a smoking cessation and/or harm reduction tool. In contrast, of the submissions from government and peak health bodies, 20 (91%) recommended against relaxing current restrictions until more evidence on efficacy and long term safety is available. Only one peak health body, the RANZCP, endorsed NVPs as a tool for harm reduction. There were commonalities among government agencies, peak health bodies and health charities in the way the problem was framed and understood, which we discussed in detail below.

### 3.1. Thresholds

Two main themes were identified in relation to thresholds: (1) the level and credibility of evidence that is needed to support changing current policy, and (2) the level of risk associated with NVPs that is considered to be acceptable.

### 3.2. Acceptable Level and Credibility of Evidence

The analyses of the submissions and public hearing transcripts revealed that the policy debates and recommendations centered on the lack of consensus regarding the current scientific evidence, and to some extent, differential interpretation of this evidence. Nearly all government and peak health bodies discussed the importance of evidence-based policy making and asserted that their policy positions were based on the current scientific literature. Those who recommended against relaxing current restrictions often cited statements from the NHMRC and WHO that advocate the principle of keeping NVPs off the market until their efficacy and long term safety profile are well established.

“The World Health Organization (WHO) does not currently consider e-cigarettes to be a legitimate tobacco cessation therapy.”Australian Council on Smoking and Health (Written submission 285)

At times, the PHE evidence review was cited by those who advocated for a ‘precautionary approach’ but only to criticize and/or discredit the ‘95% safer than smoking’ conclusion, claiming that the procedures followed to reach the conclusion were unscientific.

“Claims that e-cigarettes are “95% safer” than tobacco smoke, however, are unfounded and devoid of any scientific basis.”Cancer Council Australia and National Lung Foundation of Australia (Written submission 295 page 5)

A key argument for retaining current policy was a perceived lack of clear and convincing evidence regarding the safety and efficacy of NVPs. However, the need for more evidence to determine potential long-term health effects was universally acknowledged, including among those advocating policy change.

“The college acknowledges that further research is required to ascertain the effectiveness of e-cigarettes and vaporizers as tools for smoking cessation and the extent of harm associated with e-cigarettes and vaporizers”.Dr Shalini Arunogiri (RANZCP) (public hearing; 8 September 2017; page 6)

While the statement that “further research is needed” was common, and many stated that they would support the use of NVPs in the future if they are proven to be safe and effective, most did not elaborate on what level of safety and effectiveness, or what type of evidence would provide sufficient proof. The Thoracic Society of Australia and New Zealand (TSANZ) indicated in the public hearings that an evidence threshold of greater than 10 years of epidemiological data was required.

“Chronic respiratory conditions can take many years to become symptomatic. It is therefore important studies the track health impacts over the long term, by which we mean greater than 10 years”.Professor Bruce Thompson (TSANZ) (public hearing; 5 October 2017; page 1)

Another finding was the inconsistent use of anecdotal evidence. Case reports, social media stories and personal experiences that reported nicotine poisoning and battery explosions were cited as evidence that NVPs posed an unacceptable risk. However, similar anecdotal experiences of smokers who successfully quit smoking by switching to vaping were depicted as unreliable, and it was argued that they should not be taken into consideration in the policy making process.

“I also want to briefly highlight the safety concerns about batteries used in e-cigarettes, with reports of these devices exploding and causing quite serious injuries.”Dr Tony Bartone (AMA) (public hearing; 5 October 2017; page 9)

“Positive personal testimonies represent flagrant self-selection bias about success and cannot be given any credibility when it comes to making generalizations about the success or otherwise of a cessation method.”Professor Simon Chapman, Professor Mike Daube, David Bareham, and Associate Professor Matthew Peters (Joint submission 313)

### 3.3. Acceptable Level of Risk

Discussion about safety by most health organizations and government bodies emphasized the absolute safety that comes from abstinence rather than reducing harm via substituting smoking with a less harmful product. In this regard, the acceptable level of safety of NVPs was framed by the absolute term ‘safe’, rather than the relative term ‘safer’. For example, when asked about the threshold at which safety of NVPs would become acceptable, the AMA’s Dr Bartone said “*I’m not going to be tied to a number other than 100 per cent*” (public hearing; 5 October 2017; page 12)

“The only way to absolutely reduce the risk is not to ingest any nicotine or any of the ingredients of either e-cigarettes or tobacco.”Associate Professor John Litt (RACGP) (public hearing; 5 October 2017; page 21)

“Evidence shows there are only two effective ways to minimize the long-term harms of smoking—to quit or to avoid take-up”.Cancer Council Australia and National Heart Foundation of Australia (Written submission 295)

In contrast, the majority of submissions from HCPs and public health academics (83%) emphasized the notion that NVPs are much lower risk than combustible cigarettes, and that switching from smoking to vaping can improve overall health compared to continuing to smoke.

“Put simply, for those who value smoking or otherwise find it difficult to quit, the switch to a viable substitute is a far easier option than quitting all nicotine, something requiring sustained self-control”.Professor Ron Borland (written submission 216)

None of the health organizations or government health bodies argued that vaping is *more* harmful than smoking, but making combustible cigarettes a point of reference against which the safety of other nicotine containing products are compared was portrayed as inappropriate. The discussion hinged on the potential harms relative to abstinence, and the difficulty of inferring the degree of harm reduction that could be achieved from switching between products based on the current scientific evidence.

“While e-cigarettes may expose users to fewer toxic chemicals than, say, a tobacco cigarette, the extent to which it reduces harm has actually not been determined by the evidence.”Ms Samantha Robertson (NHMRC) (public hearing; 8 September 2017; page 18)

“What I find curious is that cigarettes kill people. That’s it. If that’s your baseline then that’s a really interesting baseline to work from. You’re basically saying, ‘Okay, we’re not going to kill you, but we’re going to do significant harm to you instead.’ I have a problem with that”Professor Bruce Thompson (TSANZ) (public hearing; 5 October 2017; page 2)

“I can’t see anything that would really encourage me to use something that I know is still going to cause harm. That is really the bottom line. At this stage, we don’t have any evidence to say that it causes zero harm.”Dr Tony Bartone (AMA) (public hearing; 5 October 2017; page 10)

This was further elaborated in the statement made by the TSANZ which accepted the notion that NVPs might save lives that otherwise would have been lost due to combustible cigarettes, but at the same time argued that those smokers who switched would “*be harmed and crippled with respiratory conditions, sitting on oxygen and in beds all the time.*” (TSANZ, public hearing 5 October 2017; page 2). No evidence was cited, such as published studies, to support the assertion that vaping results in these outcomes.

### 3.4. Approaches to NVP Regulation

We identified five common themes in the submission and hearing discussions about the recommended NVP regulations: (1) Policy coherence, (2) maintaining the status quo: the precautionary approach, (3) therapeutic product regulation: medicinal licensing and prescription access, (4) consumer product regulation: a risk proportionate regulatory model, and (5) incorporating NVPs into existing tobacco control regulations. Arguments made for and against adopting the various regulatory approaches for NVPs and example quotes are summarized in Table 2 and Table 3.

### 3.5. Policy Coherence

Many organizations stated that Australia’s current regulatory approach to nicotine and NVPs is underpinned by various national and international health policy frameworks (such as the National Drug Strategy 2017–2026, the National Tobacco Strategy 2012–2018 and the WHO FCTC) and takes Australia’s national circumstances into consideration, particularly the significant gains made in reducing smoking rates. It was argued that maintaining current restrictions on nicotine was consistent with Australia’s ‘harm minimization’ principle. They argued that advocates of NVPs focused only on one of the pillars (harm reduction) of harm minimization, while rejecting the other two important pillars (demand and supply reduction).

“Many advocates of e-cigarettes as a harm reduction approach seem to wish to disregard the other important pillars that ensure the minimizing of harm. Supply reduction principles would support regulation of supply to limit the availability of e-cigarettes”Public Health Association of Australia (Written Submission 301)

Many submissions also appeared to refer to the WHO statement, which recommends that countries with relatively low smoking rates should not open up the market for NVPs because it will not significantly reduce mortality and morbidity *“even if the full theoretical risk reduction potential of [e-cigarettes] were to be realized”* [18].

“Australia’s extraordinarily low smoking prevalence, particularly among younger people, adds significant weight to the WHO advice in a domestic context.”Cancer Council Australia (CCA) and National Heart Foundation of Australia (Written submission 295)

In contrast, other submissions asserted that the current regulatory framework governing NVPs is incoherent in that it exempts tobacco cigarettes, the most dangerous nicotine delivery system, from the same controls that are being applied to NVPs containing nicotine. The current policy was also described as more restrictive than medical use of cannabis, an internationally controlled substance.

“We are treating them much like we do heroin. It’s a much more restrictive policy towards nicotine products, for example, than medical cannabis, which the TGA is regulating in a much more liberal way at the moment, in the absence of evidence of efficacy.”Professor Wayne Hall (public hearing; 8 September 2017’ page 7)

Generally, the need for developing a national policy framework while maintaining consistency with jurisdictional legislation and coherence with other national health policy frameworks (such as the National Tobacco Strategy) was frequently discussed.

### 3.6. Maintaining the Status Quo: The Precautionary Approach

The majority of peak health organizations and government bodies described the current regulatory controls as a “precautionary approach” or as following the “precautionary principle”. This terminology was used in submissions from all government health bodies. Many policy actors (including health charities such as NHFA and ACOSH) also endorsed and/or referred to statements made by organizations that reference the precautionary principle/approach (such as NHMRC) or followed a regulatory approach with a similar sentiment to the precautionary principle (for instance, the Royal Australasian College of Surgeons and VicHealth).

“Only once safety and efficacy has been thoroughly established should consideration about changing regulatory approaches take place”.AMA (Written submission 289)

The term ‘precautionary’ was frequently used, but there appeared to be variations in what this meant in practical terms for NVP regulation. The first and most common variation of this approach was that the introduction of any new product that carries unknown but potentially harmful effects should be prohibited until the potential harms of the product, and their efficacy, is established with adequate evidence. Those who advocated for a precautionary approach reiterated the notion that any change to the *status quo* should only be considered after it is determined, with sufficient certainty, that NVPs can help people quit and also that they are ‘safe’.

“The longitudinal research that is required to establish safety will take time, but until more definitive evidence on safety becomes available the precautionary principle should be applied to these products”.AMA (Written submission 289)

As described above, some believed that adequate evidence would not become available for decades, and some suggested that the evidence had to show zero harm. The burden of proof that it is safe and effective was argued to fall to NVP advocates and manufacturers who want to see a change in the current regulatory landscape.

“If there is a suspected risk of harm and the scientific information is lacking, such that there is an absence of scientific consensus, then the burden of proof that it is not harmful falls on those wanting to progress the issue”.TSANZ and Lung Foundation (written submission 332)

The second argument in favor of adopting a “precautionary approach” was that using nicotine in any form (be it in cigarettes or NVPs) other than approved nicotine replacements (NRTs) was extremely dangerous and addictive. It was stated that NRTs do not predispose or worsen users’ addiction to nicotine (due to non-addictive concentrations and slower release of nicotine) and they are not being taken up by non-smokers even though they are available in general retail outlets. The distinctive feature of NVPs, however, posed an ‘increased risk’ for users, by sustaining addiction to nicotine and potentially deterring them from quitting altogether, i.e., becoming long-term dual use of cigarettes and NVPs.

“E-cigarettes may result in some smokers delaying their decision to quit, as people may feasibly move between e-cigarettes and tobacco smoking, as their desire to quit varies over time.”AMA (Written submission 289)

“What the patches and gums do is provide a low-level sustained response. Why e-cigarettes are indeed popular is that, in the same way a heroin injection gives a hit, you get a large amount of nicotine coming into the system. It gives you a nicotine hit. That’s why there is a risk of diversion, use—call it what you want—for non-smoking cessation purposes.”Dr John Skerritt (TGA, DoH) (public hearing; 8 September 2017; page 13)

### 3.7. Therapeutic Products Regulation: Medicinal Licensing and Prescription Access

Part of the policy debate hinged on whether or not NVPs should continue to be regulated as a medicine due to the implications for licensing, production and supply. Those who opposed relaxing current restrictions asserted that any nicotine-containing products that are inhaled directly into the lungs, be it inhalers or NVPs, should be regulated as prescription medicines and that they should be assessed, approved and regulated by the TGA regardless of the claims made (smoking cessation or harm reduction). ACOSH’s submission emphasized that “*no action should be taken that in any way pre-empts the role of the TGA*” (Written Submission 285, page 2). It was reiterated that any product with a harm reduction claim should go through TGA approval process regardless of its potential benefit in helping smokers quit because *‘they are being framed as a health argument’* (Mr Michael Moore [PHAA]; public hearing; 5 October 2017; page 17). It was argued that unless they are approved by the TGA as a cessation aid, NVPs should neither be marketed, nor legally available as quit aids.

TGA approval and prescription only access was described as a *‘sensible harm-reduction approach’* (*Professor Bruce Thompson [TSANZ]; public hearing; 5 October 2017; page 2*) and advocated as a solution to avoid children taking up NVPs and eventually transferring to smoking. The TGA also cited potential benefits of the current approach. It was argued that making NVPs available only on prescription would benefit priority populations as these patients could use routine GP visits to collect their vaping products alongside other medications. If approved vaping products were listed on the Pharmaceutical Benefits Scheme, this would make them available at ‘concessional rates’.

The current legal framework allows smokers to legally access nicotine e-liquid either through TGA exemptions (personal importation, the Special Access Scheme, Authorized Prescriber Scheme) or through extemporaneous compounding. These legal pathways all require the person to hold a valid medical prescription. Yet, there appeared to be misunderstanding of the Commonwealth law among some government health bodies and health charities. For instance, RACGP argued that due to the extensive and ‘onerous’ procedure of writing a prescription for nicotine under the current regulatory arrangements, GPs would be less likely to engage in such process, and this would ‘*make it extremely difficult for people who want to get access to some form of nicotine and import it*’ Associate Professor John Litt [RACGP]; public hearing; 5 October 2017; page 20). However, while the Special Access and Authorized Prescriber Schemes do involve considerable additional paperwork for prescribers, the personal importation and extemporaneous compounding options do not require any special paperwork beyond a prescription.

The issue of biomedicalization of smoking cessation as it relates to NVPs and medicinal regulation was also discussed by the RANZCP. It was noted that smokers may not opt to use prescription medications for smoking cessation since smoking is not widely seen as a medical issue. This suggested that the therapeutic goods framework was not appropriate if NVPs were viewed by smokers as being in the same category as cigarettes, since this would exclude some who may benefit from wider access outside a therapeutic framing.

### 3.8. Consumer Product Regulation: A Risk Proportionate Regulatory Model

While the potential for NVPs to be a cessation tool was acknowledged by some, the risk of marketing these products as a consumer product was described as unacceptable due to the perceived potential risks of these products prevailing over the unproven health claims. Others argued that treating NVPs as a consumer product is ‘harm escalation’ rather than ‘harm reduction’, presumably because the products would primarily be used for purposes other than quitting smoking if not regulated as therapeutic goods. The RANZCP was the only peak health body represented in the submissions and hearings that advocated for regulating NVPs as a consumer product. To reconcile the benefit of using NVPs for priority populations (such as those with mental illness) with the potential risk of young people taking up smoking, the RANZCP suggested following a risk proportionate regulatory model that would discourage use of smoked tobacco products, and allow appropriate access to NVPs among smokers while discouraging access to and use by adolescents. In this context, considering NVPs as tobacco products disregards the differential risks associated with vaping and smoking while subjecting them to TGA processes (medicinal licensing) would heavily hinder access to these products. Thus, regulating NVPs as consumer products was viewed by RANZCP as an appropriate policy response.

A number of submissions from HCPs and public health academics also reinforced the notion that NVPs should be seen as a consumer product (not as a clinical intervention) and regulated as such, but within a fairly restrictive and risk proportionate manner. Moreover, RANZCP advocated for applying differential taxes for differential risks. As such, NVPs would be taxed at a much lower rate than conventional cigarettes to promote switching.

“The continuing harms to disadvantaged groups reduced most quickly, if two additional sets of strategies are integrated into the existing mix: namely, regulating smoked tobacco more stringently, and differentially regulating (including pricing) lower harm smokeless nicotine and tobacco products to make switching to these products a more viable option for those unwilling or unable to quit nicotine use completely.”Professor Ron Borland (written submission 216)

“Keeping e-cigarettes and vaporizers at a low cost would not only encourage uptake of these devices over more harmful products but would also present financial benefits for vulnerable groups.”Royal Australian and New Zealand College of Psychiatrists (written submission 294)

### 3.9. Incorporating NVPs into Existing Tobacco Control Regulations

The other regulatory option discussed frequently was treating NVPs as tobacco products particularly in terms of promotion, advertising, sale to minors and vaping in smoke-free areas. This regulatory option, which was seen as compatible with medicines regulation, was mostly discussed within the context of the precautionary principle and the importance of having a ‘controlled availability’ of products that have an unknown level of harm. Some of the proposed specific regulatory approaches are summarized in Table 4.

### 3.10. Promotion and Advertising

It was argued that the current promotion and marketing strategies used for NVPs, most of which takes place online, are intended to entice youth and have the potential to re-normalize smoking. It was reiterated throughout many of the submissions that NVP companies are strategically targeting youth and young adults with the use of social media as advertising channels, glamourizing their products and using flavors that appeal to young people.

“Of significant concern is that this [e-cigarette] marketing is now directed at young people with the use of flavorings such as strawberry custard, chocolate and cereal, amongst others”.Thoracic Society of Australia and New Zealand and Lung Foundation Australia (written submission 332)

Some submissions suggested that the marketing and advertising of NVPs should be subjected to the same restrictions as tobacco cigarettes, as set out in the Tobacco Advertising Prohibition Act 1992. This includes prohibiting all types of promotion, advertising, free distribution, sponsorship and point-of-sale display of vaping products. On the other hand, RANZCP recommended heavily restricting advertising instead of complete prohibition, as this would balance the benefits of raising awareness on the relative harms of vaping for current smokers with reducing appeal to youth and non-smokers.

“In the RANZCP’s view, severe restrictions may be preferable to complete prohibition insofar as responsible advertising may raise awareness of the benefits of these products over their more harmful alternatives.”Royal Australian and New Zealand College of Psychiatrists (written submission 294)

Tobacco industry involvement in the NVP market, the unsubstantiated and misleading claims made by NVP companies regarding safety and efficacy, youth-targeted marketing (including use of sweet flavorings), and the difficulty of monitoring internet-based advertisements were flagged as reasons why the government should completely ban or heavily restrict vaping product promotion and marketing activities.

“My perspective is that big tobacco is attempting to reframe the electronic cigarette issue as a health issue firstly, and they are also attempting to redefine what we mean by a tobacco-free world.”Mr Michael Moore (PHAA) (public hearing; 5 October 2017; page 14)

“This emphasis on online marketing can make monitoring and policing the claims made by online e-cigarette retailers difficult… and there is a real risk that consumers will continue to be misled about the safety of e-cigarettes as well as their role in smoking cessation”.Australian Medical Association (written submission 289)

### 3.11. Sale and Supply to Minors

The need to prevent children and adolescents from accessing vaping products was frequently discussed and unanimously agreed on by all organizations and HCPs. Enacting an age restriction on sales of vaping products and prohibiting vending machine sales were some of the policy measures recommended to restrict access to minors.

“E-cigarettes and the related products should only be available to those people aged 18 years and over”.Australian Medical Association (written submission 289)

“The RANZCP would also support prohibiting the sale of e-cigarettes and vaporizers in vending machines which, if allowed, might facilitate access for under-age users.”Royal Australian and New Zealand College of Psychiatrists (written submission 294).

### 3.12. Smoke-Free Policy and NVPs

It was frequently mentioned that Australia has been successful in developing a strong smoke-free environment and that access to NVPs and allowing vaping in smoke-free areas may reverse this culture and make smoking socially acceptable again. The majority of organizations, thus, advocated to prohibit vaping, with or without nicotine, in smoke-free areas. An exception to this, however, was the submission made by RANZCP, which recommended allowing vaping in smoke-free mental health facilities.

“The RANZCP also notes that many mental health facilities are now smoke-free and there may be benefits in allowing the use of e-cigarettes and vaporizers in these settings. This may encourage patients to switch to these less harmful alternatives while reducing the conflicts which smoking bans can sometimes cause.”Royal Australian and New Zealand College of Psychiatrists (written submission 294).

### 3.13. Committee Recommendations

After considering submissions, and the views expressed at public hearing, the majority of the Committee suggested in their report that the current regulatory arrangements and exemptions for nicotine and nicotine containing NVPs (including the role of the TGA) remained appropriate and indicated the need for: (1) developing a nationally consistent regulatory approach for non-nicotine containing NVPs; (2) conducting ‘an independent and comprehensive review of the evidence’ regarding vaping; and (3) developing policy frameworks for assessing and restricting various flavorings used in vaping products. A minority of the Committee, in dissenting reports, recommended that NVPs be made available as consumer products with regulation similar to what has been implemented in the UK.

## 4. Discussion

It is widely accepted among the public health community that policy making processes should be informed by data and evidence [19], although the types of evidence that are relevant will be contingent on the way the issue is framed and understood [20]. The limited, and at times, conflicting body of evidence regarding the safety and efficacy of NVPs has resulted in the emergence of profoundly divergent ideologies among policymakers and the public health community [21].

In this study, we examined how health policy actors in Australia framed NVP-related debates and how evidence is taken up and interpreted to justify the different concerns and proposed policy priorities. Among the possible regulatory responses, recommendations against relaxing current restrictions on NVPs (maintaining the status quo) had a greater consensus among government regulators, peak health bodies and health charities. Most policy actors advocated for abstinence as the only acceptable goal for smokers, emphasized the potential risks associated with vaping products and called for extreme caution in the use and marketing of these products. A detailed analysis of submissions and interview transcripts indicates that the health-related concerns frequently expressed by policy actors who advocated for maintaining the status quo were confined largely to what is not known about NVPs rather than what is already known. There was also a widespread concern that that the potential societal harm posed by these products would outweigh the ‘unproven’ potential for any individual benefit. On the other hand, the majority of individual HCPs and public health academics who made submissions placed tobacco harm reduction at the center of their policy objectives and characterized NVPs as novel nicotine delivery technologies that might lead to a dramatic decline in the use of combustible cigarettes.

Although the inherent nature of risk trade-offs in NVP regulation was universally recognized, consideration and tolerability of these trade-offs appeared to vary between stakeholders. Policy actors who recommended against relaxing the current restrictions gave greater weight to potential risk to current non-smokers over potential benefit to smokers. They asserted that Australia, having a very low prevalence of smoking, should not open up the market to NVPs due to the risk of recruiting new smokers, particularly young people, suggesting that maintaining the status quo is the lower risk policy option. In contrast, those who placed tobacco harm reduction at the center of their policy objectives (RANZCP and majority of individual HCPs and public health academics) recommended a risk proportionate regulatory model that would allow access to NVPs for smokers while discouraging access to and use by adolescents. They also argued that the risks of smoking are likely to be substantially higher than vaping, vaping may also deter or divert youth from taking up smoking, and that a lack of access to an acceptable substitute product such as NVPs could plausibly perpetuate smoking. It is also noteworthy that while the need for more evidence was universally acknowledged, there were differing views on what was an adequate level of confirmatory evidence required before making any judgement or enacting a policy response. It was common for government agencies and health bodies to assert that unless the safety profile of long term use of NVPs is known, through decades of epidemiological evidence, the current regulatory approach should remain. Some also asserted that NVPs would need to be proven to be 100% risk free to be an acceptable product for smokers to use. This approach was often referred to in the submissions as a ‘precautionary approach’ or consistent with the ‘precautionary principle’ as justification for the current restrictive policy on nicotine. This interpretation of the precautionary principle differs from that of the European Union, which requires policy makers to take the ‘level of risk’ into consideration when deciding to take proportionate regulatory actions and to consider the consequences of the decision (both inaction and intervention) [22].

Despite the UK, New Zealand, and Australia having similar tobacco control policies and population smoking prevalence, the public health communities in the UK and New Zealand have come to starkly different positions compared to Australia. The majority of health policy actors in the UK have framed NVPs as a tobacco harm reduction tool and endorsed the use of NVPs as a smoking cessation aid among smokers wanting to quit, but who are unable to do so by other means [23]. This framing also appeared to have had traction in the regulatory process as evidenced by the final ruling and stances of notable medical and public health bodies [3,24]. Similar to Public Health England [3], the New Zealand Ministry of Health also encourages stop smoking services to become ‘vaping friendly’ and to support smokers to use NVPs to quit smoking [25]. Although the New Zealand government is in the process of developing new regulations for NVPs that will continue regulating them as consumer products [26], NVPs and e-liquid containing nicotine became legal to sell as a consumer product under the Smoke-free Environments Act 1990, following the outcome of a court ruling in 2018 [27]. In contrast, the outcome of the Australian parliamentary inquiry was a majority report that recommended retaining the current regulatory arrangements and to also look at introducing further restrictions [21]. This suggests that unlike the UK and New Zealand, NVP-related policy outcomes in Australia are grounded in and shaped by conventional tobacco control policies that focus on demand reduction and abstinence (such as tobacco taxation, mass media campaigns, and health warnings), rather than harm reduction approaches. The UK, on the other hand, has invested to a greater extent than Australia in smoking cessation assistance, such as dedicated stop smoking clinics, and provision of all licensed smoking cessation medicines for free. In this context, widening access to NVPs as a smoking cessation and harm reduction tool is consistent with the greater focus on smoking cessation. It also reinforces the notion that the evidence-based policymaking process involving intractable policy controversies goes beyond a dispassionate evaluation of the evidence base, and is influenced by a wide array of political and institutional factors.

Although the prevalence of daily smoking among adult Australians decreased from 24.3% in 1991 to 12.8% in 2013, there has been only minimal decline over the last few years with only 0.6% reduction rate seen between 2013 and 2016 (from 12.8% in 2013 to 12.2% in 2016) and smoking prevalence remains highly prevalent among many priority populations, including people on low incomes, Indigenous peoples and people with mental illnesses [28]. This is despite Australia being an international leader in implementing tobacco control strategies which reduce the demand and supply of tobacco products. Although the National Tobacco Strategy 2012–2018 mentioned reducing harm from tobacco use as one of its three pillars [29], harm reduction strategies have not been widely implemented. Although NVPs are not completely safe, evidence suggests that they are less harmful than combustible cigarettes [3,30]. Thus, controlled availability of NVPs could provide a harm reduction tool for unwilling or hard-to-quit smokers. Access to a lower risk substitute for cigarettes could complement demand and supply reduction strategies. However, Australia’s current regulatory framework, however, puts severe restrictions on access to NVPs, with potentially heavy penalties for possession and use without a prescription, while allowing nicotine in the form of smoked tobacco products to be sold as consumer products. One potential consequence of maintaining greater restrictions on nicotine in less harmful forms compared to cigarettes, is that some smokers will continue smoking rather than switching to the lower risk option. Hence, overly cautious regulation also carries risks such as potentially perpetuating smoking [31], and potential adverse impacts of the current approach were seldom addressed by those referencing the precautionary principle. Moreover, the current regulatory framework does not seem to give due consideration to the views of consumers who are using or want to use NVPs as a cessation or harm reduction tool. A number of studies have reported that consumers prefer regulatory options that will not impede their ability to obtain and use NVPs [32,33,34,35]. Further research that analyses the submissions to public inquiries made by consumers and consumer organizations would be worthwhile to capture their views and preferences for regulations.

The Parliamentary submissions (and public hearing transcripts) represent the most recent and comprehensive data from Australian health organizations and health professionals on their views regarding NVP regulation in Australia. However, our study has some limitations that should be taken into account while interpreting the findings. First, the data is not nationally representative as it is a self-selected sample consisting of only those organizations who made a submission to the Inquiry. For instance, there were no submissions provided by any of the professional or peak organizations representing pharmacists in Australia. Second, submissions by public health academics and healthcare professionals were made in their individual capacity and views of peak health bodies may not represent the views of all their members. Thirdly, this study should be regarded as a snapshot in time, as the science surrounding NVPs is fast moving. For example, declines in smoking prevalence in the UK and USA, including among youth, have continued while vaping prevalence has increased [36], which may allay fears that access to NVPs containing nicotine will lead to smokers delaying quitting smoking, an increase in youth smoking via a ‘gateway’ effect, and the renormalization of smoking as a socially acceptable behavior. Furthermore, stances and recommendations of health organizations may have changed since the Inquiry was held. For instance, the Royal Australasian College of Physicians have since released a more nuanced position statement (May 2018) [37]. Nevertheless, this analysis has provided an insight into the views of Australian government and non-government health organizations and other health professionals on safety and efficacy of NVPs and how they should be regulated.

## 5. Conclusions

The majority of submissions from Australian government and peak health bodies to a Federal Parliamentary Inquiry into vaping products recommended against relaxing current restrictions on the regulation of nicotine for use in these products, describing Australia’s current regulatory landscape as a ‘sensible harm-reduction approach’. In contrast, most of the submissions from HCPs and public health academics endorsed the role of NVPs as a smoking cessation and/or harm reduction tool. A key argument for prohibiting the sale of nicotine containing vaping products was a perceived lack of clear and convincing evidence regarding the efficacy of NVPs and the difficulty of inferring the likely health outcomes for smokers who switched to vaping based on the current scientific evidence. Overall, the health-related concerns frequently expressed in the submissions focused largely on the unknown and theoretical risks related to NVP use. There was widespread support for age restrictions on sales and advertising and promotion restrictions, with most submissions supporting similar controls as for tobacco products. A number of barriers to widening access to nicotine for use in NVPs were evident, including the demand for long-term evidence that confirms an unusually high standard of safety and efficacy.

## Figures and Tables

**Table 1 ijerph-16-04555-t001:** Submissions included in the analysis, *n* = 40.

Government Bodies, *n* = 8
1. Department of Health (DoH)
2. National Health and Medical Research Council (NHMRC)
3. New South Wales Health *
4. South Australian (SA) Government *
5. Queensland Government *
6. Western Australia (WA) Government *
7. Tasmanian Government *
8. VicHealth *^,◊^
**Peak Health Bodies, *n* = 7**
1. Australian Dental Association *
2. Australian Medical Association (AMA)
3. Royal Australian College of General Practitioners (RACGP)
4. Royal Australasian College of Physicians (RACP) *
5. Royal Australasian College of Surgeons (RACS) *
6. Royal Australian and New Zealand College of Psychiatrists (RANZCP)
7. Thoracic Society of Australia and New Zealand (TSANZ)
**Health Charities, *n* = 7**
1. Australian Council on Smoking and Health (ACOSH)
2. Cancer Council Australia (CCA)
3. National Heart Foundation of Australia (NHFA)
4. National Heart Foundation of Australia, WA Division
5. Public Health Association of Australia (PHAA)
6. Quit Victoria *
7. Lung Foundation Australia
**Individual Healthcare Professionals (HCPs) and Academicians *, *n* = 18**
1. Addiction medicine specialists (*n* = 4)
2. General practitioners (GPs) (*n* = 2)
3. University public health academics (*n* = 12)

* Did not participate in a public hearing. ^◊^ Health Promotion Foundation funded by the Victorian Government.

**Table 2 ijerph-16-04555-t002:** Arguments made for adopting various regulatory approaches for NVPs and example quotes.

Regulatory Approach *	Advocate for Adopting the Approach		
	Key Arguments	Example Quotes	Example Submissions
Medicinal regulation	(1) Since NVPs are advertised as a quit aid, they should be subject to therapeutic regulation and licensed as medicines; (2) Protects public health by ensuring maximum safety and efficacy; (3) Minimizes risk of uptake by unintended population (children and non-smokers) and eventually transferring to smoking; (4) Would make NVPs easily available at ‘concessional rates’ via the Pharmaceutical Benefits Scheme	“Especially for groups who, for example, are seeing their GP or psychiatrist reasonably regularly, an avenue to prescription access to these products would be potentially an attractive option”. Dr John Skerritt (TGA, DoH) (public hearing; 8 September 2017; page 17)	AMA, RACGP, RACP, RACS, TSANZ, ACOSH, CCA, NHFA, PHAA, Quit Victoria, LFA, and all government bodies
Consumer product regulation	(1) NVPs are consumer goods designed to replace an existing, more harmful consumer product; (2) It will ensure general safety and allow them to be regulated proportionate to their risks; (3) Allows product improvement and innovation.	“E-cigarettes and vaporizers should be treated as consumer products, not tobacco products or medicines. They should be controlled proportionate to their risks, whilst still allowing for individuals to have appropriate access to these products.” Dr Shalini Arunogiri (RANZCP) (public hearing; 8 September 2017; page 6)	RANZCP and 15 (out of 18) HCPs and academics
Tobacco product regulation	(1) Is an effective demand reduction strategy; (2) Makes it difficult for the tobacco industry to market NVPs to young people; (3) Subjecting promotion and advertising of NVPs to tobacco product regulation would prevent the unsubstantiated claims and youth-targeted marketing.	“We, therefore, strongly recommend that the use of e-cigarettes be prohibited in legislated smoke free areas in all Australian jurisdictions (even if ultimately approved by the TGA for therapeutic use).” Australian Council on Smoking and Health (written submission 285)	AMA, RACP, RACS, TSANZ, ACOSH, CCA, NHFA, PHAA, Quit Victoria, LFA, and all government bodies

* A dual regulatory pathway (regulating NVPs as consumer products while maintaining medicines regulation for NVPs that claim therapeutic benefits) was indicated as an alternative approach in some of the individual submissions (Written submissions 258, and 282).

**Table 3 ijerph-16-04555-t003:** Arguments made against adopting various regulatory approaches for NVPs and example quotes.

Regulatory Approach	Advocate Against Adopting the Approach		
	Key Arguments	Example Quotes	Example Submissions
Medicinal regulation	(1) NVPs are consumer driven products being used as safer alternatives to tobacco products, not as medicines; (2) The onerous and costly applications to the TGA and compliance for each product creates substantial barriers to entry and hinders innovation; (3) Increases cost due to doctor visits and pharmacy charges, making them a less attractive option for smokers compared to cigarettes.	“A broader sociocultural question around how smokers see their smoking, and people who currently smoke cigarettes, who may not necessarily see it as an illness for which they need to go and get medication. So there may be another population that may not actually engage in that process.” Dr Shalini Arunogiri (RANZCP) (public hearing; 8 September 2017; Page 29)	RANZCP, and individual submissions (Submission number 258, 282, 216)
Consumer product regulation	(1) Presence of tobacco cigarettes as consumer products ‘does not provide a reasonable basis to expose the public’ to other products such as NVPs; (2) It is a ‘harm escalation’ rather than ‘harm reduction’ approach; (3) Should not be consumer products since they are being framed as a health argument; (4) Will exacerbate the aggressive NVP marketing and ‘drive take-up rather than confer a health benefit’.	“Making these products freely available as a consumer good, when it is a product that causes damage to the lung, is not harm reduction. Indeed, it is indeed harm escalation.” Professor Bruce Thompson (TSANZ) (public hearing; 5 October 2017; page 2).	Department of Health, LFA, TSANZ, PHAA, CCA, NHFA
Tobacco product regulation	(1) Should not be treated as tobacco products since NVPs do not contain tobacco and do not combust; (2) This approach would carry a misleading or inaccurate message that NVPs are equally harmful as tobacco cigarettes; (3) Reducing smokers’ use of NVPs is against the evidence that encouraging increased NVPs use substantially reduces tobacco-related harm.	“It is an incoherent public health policy in that it bans the sale of less harmful nicotine products while allowing the most harmful—combustible cigarettes—to be freely sold”. Associate Professor Coral Gartner and Professor Wayne Hall (written submission 282)	RANZCP, and individual submissions (Submission number 258, 282, 216)

**Table 4 ijerph-16-04555-t004:** Proposed specific regulatory approaches to reduce the risk and maximize benefits of NVPs.

Regulatory Approach	Recommendations	Example Submissions
Restrictions on sale	Sale only to people aged 18 or over	AMA, ADA, RANZCP, RACP, RACS, TSANZ, ACOSH, CCA, NHFA, PHAA, Quit Victoria, LFA, all government bodies and individual submissions (Submission number 112, 164, 258, 282, 216)
	Restrict sale to only specialist vape shops, tobacconists and adult stores	Submission number 282
	Prohibit sale in vending machines	RACS, RANZCP, Submission number 258
Vaping in smoke-free areas	Prohibit in all areas that are designated to be smoke-free	RACP, ACOSH, PHAA, RACS, TSANZ, LFA, VicHealth, Submission number 313, and all government bodies
	Prohibit indoor public use	TSANZ, LFA, Submission number 282
	Allow businesses and local authorities to make their own decisions	Submission numbers 258, and 216
	Allow vaping in some smoke-free places, such as mental health facilities	RANZCP
Quality standards and device safety	Subject to the Australian Competition and Consumer Commission (ACCC)	RACP, Submission number 282
	Require disclosure, testing and monitoring of product composition	RANZCP
Advertising & promotion	Subject to the same restrictions as tobacco cigarettes	AMA, ADA, RACP, RACS, TSANZ, ACOSH, CCA, NHFA, PHAA, Quit Victoria, LFA, Submission number 313
	Limited forms of promotion directed primarily at smokers to encourage switching	Submission number 216
	Restrict to point of sale	Submission number 282
Flavor restriction	Allow flavors with exception of those with known adverse effects	Submission number 216, 258
	Restrict flavors that appeal to children and young people	RACS, Submission number 216
Taxation	Subject to excise tax, at a lower rate than that of tobacco cigarettes	RACP, RANZCP, Submission number 216, 258
Packaging and labelling	Require child resistant closures	TSANZ, LFA, RACP, RACS, Submission number 282
	Require listing of all ingredients and safety instructions or health warnings	TSANZ, LFA, RACP, RACS, RANZCP, Submission number 258

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
