# Peer review of "Policy Debates Regarding Nicotine Vaping Products in Australia: A Qualitative Analysis of Submissions to a Government Inquiry from Health and Medical Organisations"

_ijerph, 2019, doi:10.3390/ijerph16224555_

Round 1

Reviewer 1 Report

I read the manuscript with interest.

My main concern relates to the different categories of stakeholders and how they are juxtaposed, particularly in the discussion and when drawing conclusions. It is unclear how the categories of stakeholders were decided upon. This is important because the analysis builds on these categories. I wondered, for example, why bodies representing health professionals, e.g. the Royal Australian College of General Practitioners/Physicians/Surgeons etc, were not included in the category of Healthcare Professionals (HCPs) and academics but are a separate category. Also, the findings do not make it particularly clear that different groups took different positions. In fact, the authors present the themes and highlight often quite similar quotes from different types of organisations.

The introduction, discussion and conclusion do not align. Differences between NZ and AUS are not introduced in the introduction but discussed in length in the discussion. The main point of the discussion is not clear. The conclusion about different stakeholders taking different positions does not seem to be grounded in the findings.

As the authors say, the study "has provided an insight into the views of Australian government and non-government health organisations and other health professionals on safety and efficacy of NVPs and how they should be regulated". The current manuscript, however, lacks depth of analysis. While the findings are presented in a concise way, the discussion does not go into detail and depth and remains rather shallow and does not go beyond descriptive analysis. I suggest that the authors are encouraged to tease out the particularities of the Australian context in comparison to other nations (which might increase the relevance of this paper for tobacco control internationally) and to reflect on the particular approach to NVP regulation in AUS. In order to do this, some deeper discussion of tobacco control in AUS might be necessary to place recent developments on NVPs in a broader context.

The authors might want to draw on the paper published recently in IJERPH on e-cigarette policy debates in Scotland to compare the engagement of health-related stakeholders in the AUS context to that of similar stakeholders in other nations.

Author Response

Reviewer 1

I read the manuscript with interest.

My main concern relates to the different categories of stakeholders and how they are juxtaposed, particularly in the discussion and when drawing conclusions. It is unclear how the categories of stakeholders were decided upon. This is important because the analysis builds on these categories. I wondered, for example, why bodies representing health professionals, e.g. the Royal Australian College of General Practitioners/Physicians/Surgeons etc, were not included in the category of Healthcare Professionals (HCPs) and academics but are a separate category.

Response: Stakeholder submissions were classified as being from an organisation or from an individual health practitioner or academic. This is because the positions of peak health organisations do not necessarily represent the views of particular individuals within that organisation. Positions of peak health bodies are more likely to influence policy than the views of individual healthcare practitioners, however it is still important to consider the views of individual healthcare practitioners. It was clear from our analysis that of the healthcare professionals who made their own individual submissions, most were at odds with the positions of the peak health bodies. Here are some example quotes.

“The longitudinal research that is required to establish safety will take time, but until more definitive evidence on safety becomes available the precautionary principle should be applied to these products” Australian Medical Association (Written submission 289)

“We are treating them much like we do heroin. It's a much more restrictive policy towards nicotine products, for example, than medical cannabis, which the TGA is regulating in a much more liberal way at the moment, in the absence of evidence of efficacy.” Professor Wayne Hall (public hearing; 8 September 2017’ page 7)

Public health academics who made submissions were also considered as individual submissions, because they did not state they were making their submissions on behalf of their organisation (i.e., their university).

Also, the findings do not make it particularly clear that different groups took different positions. In fact, the authors present the themes and highlight often quite similar quotes from different types of organisations.

Response: As clearly stated in the findings, all (except RANZCP) health bodies and government agencies have the same position (supporting the status quo). The difference, however, lies in the way some arguments are presented, particularly on safety issues and level of evidence. These commonalities among government agencies, peak health bodies and health charities in the way they framed and understood the problem is now discussed at length in the revised manuscript.

The introduction, discussion and conclusion do not align. Differences between NZ and AUS are not introduced in the introduction but discussed in length in the discussion. The conclusion about different stakeholders taking different positions does not seem to be grounded in the findings.

Response: We have revised the introduction section to be in line with the discussion and conclusion. Differences between regulatory approaches of UK and NZ (vs Australia) is now briefly mentioned in the introduction.

“Government agencies in some countries such as the UK and New Zealand have not only endorsed a role for NVPs in reducing tobacco related harm, but are also proactively encouraging smokers to switch to vaping, including supporting stop smoking services to become ‘vaping friendly’. In the UK, a dual regulatory pathway exists for NVPs as either a medicine or a consumer product under the EU Tobacco Product Directive (TPD). Similarly, NVPs can be marketed as either medicines, if health claims are made, or tobacco products in New Zealand following the outcome of a court case in 2018.” Please see page 1-2, lines 40-47.

Our conclusion that “the majority of submissions from Australian government and peak health bodies recommended against relaxing current restrictions while most of the submissions from HCPs and public health academics endorsed the role of NVPs as a smoking cessation and/or harm reduction tool” is grounded in and supported by the findings. Please see page 17, lines 623-631.

“The majority of submissions from Australian government and peak health bodies to a Federal Parliamentary Inquiry into vaping products recommended against relaxing current restrictions on the regulation of nicotine for use in these products, describing Australia’s current regulatory landscape as a ‘sensible harm-reduction approach’. In contrast, most of the submissions from HCPs and public health academics endorsed the role of NVPs as a smoking cessation and/or harm reduction tool.”

The main point of the discussion is not clear. As the authors say, the study "has provided an insight into the views of Australian government and non-government health organisations and other health professionals on safety and efficacy of NVPs and how they should be regulated". The current manuscript, however, lacks depth of analysis. While the findings are presented in a concise way, the discussion does not go into detail and depth and remains rather shallow and does not go beyond descriptive analysis. I suggest that the authors are encouraged to tease out the particularities of the Australian context in comparison to other nations (which might increase the relevance of this paper for tobacco control internationally) and to reflect on the particular approach to NVP regulation in AUS. In order to do this, some deeper discussion of tobacco control in AUS might be necessary to place recent developments on NVPs in a broader context. The authors might want to draw on the paper published recently in IJERPH on e-cigarette policy debates in Scotland to compare the engagement of health-related stakeholders in the AUS context to that of similar stakeholders in other nations.

Response: The discussion section is now entirely rewritten in light of the comments and suggestions provided, and we have provided a detailed analysis of findings. The changes made are indicated using track changes. Generally, the particularity of Australian regulatory approach compared to other developed countries is discussed at length by:

Pinpointing how the different framings around NVPs (among policy actors in the UK and Australia) have resulted in differential interpretation of what and how much evidence is needed, and the variation in consideration to and tolerability of various risk trade-offs associated with allowing or banning NVPs. Highlighting Australia’s emphasis in conventional tobacco control policies to motivate quitting (e.g. mass media campaigns, taxation) in contrast to countries such as the UK that has invested to a greater extent than Australia in smoking cessation assistance such as stop smoking clinics. Highlighting how Australia’s highly restrictive regulatory framework (compared to the UK, New Zealand that have similar population smoking prevalence to that of Australia) could result in various unintended consequences.

We have also discussed the complementarity of tobacco harm reduction to Australia’s current tobacco control strategies, which focused largely on reducing demand and supply of tobacco. Finally, we have discussed the implication of considering the views and policy support of consumers (who are using or want to use NVPs) into consideration while enacting policy responses.

Reviewer 2 Report

This is a very nicely written qualitative analysis of arguments made in support of or against loosing restrictions on the sale and marketing of nicotine vaping products in Australia. I found the analysis informative in helping to understand arguments from different medical and public health organizations in Australia and I believe these arguments also apply in other high income countries. I only have a few minor recommendations to the authors to assist them in strengthening what I consider to be a nicely presented paper.

First, main themes presented in the text do not match with summary table 1.  Much of the material covered in the results in text could be more conveniently summarized in text since examples (quotes) are used to illustrate divergent viewpoints from those advocating for or opposing a loosing the existing restrictive regulations on nicotine vaping in Australia. Thus, I would recommend the authors shorten the text, and create a new table 1 using the themes highlighted in their paper. 

Second, the question of which organizations/groups tended to favor and/or oppose loosing the regulations on nicotine vaping should be presented towards the end or at the beginning of the article.  This information is summarized in table 2.  Both Table 1 and 2 ought to be referenced in the text. 

Third, the authors might want to comment on areas that might not have been covered well.  For example, it seems likely there was acceptance of the success of existing regulation which has yielded slow but steady progress in reducing smoking rates, but with little emphasis given to the relatively slow progress that has been made and urgency to change the slow trajectory of change predicted.  In other words, it appears there that the status quo is acceptable.

Forth, there appears to have been no discussion of how accepting a smoking harm reduction might work synergistically  with the well established demand reducing policies already implemented in in Australia.

Fifth, the debate about loosing regulations affecting access to nicotine vaping products in Australia appears not to consider the wishes of current smokers.  It would be helpful to cite polling data reflective of consumer attitudes about regulating vaping in Australia.

Finally, on a minor issue, I found the term "peak" health bodies unfamiliar.  It would help the reader to have a definition of what the authors mean by peak  health body since this term is frequently used in the manuscript.

Author Response

Reviewer 2

This is a very nicely written qualitative analysis of arguments made in support of or against loosing restrictions on the sale and marketing of nicotine vaping products in Australia. I found the analysis informative in helping to understand arguments from different medical and public health organizations in Australia and I believe these arguments also apply in other high income countries. I only have a few minor recommendations to the authors to assist them in strengthening what I consider to be a nicely presented paper.

First, main themes presented in the text do not match with summary table 1. Much of the material covered in the results in text could be more conveniently summarized in text since examples (quotes) are used to illustrate divergent viewpoints from those advocating for or opposing a loosing the existing restrictive regulations on nicotine vaping in Australia. Thus, I would recommend the authors shorten the text, and create a new table 1 using the themes highlighted in their paper.

Response: We have split Table 1 into two tables, included example quotes and incorporated within the article.

Second, the question of which organizations/groups tended to favor and/or oppose loosing the regulations on nicotine vaping should be presented towards the end or at the beginning of the article. This information is summarized in table 2. Both Table 1 and 2 ought to be referenced in the text.

Response: A paragraph on policy actors who support and oppose widening access to NVPs is presented at the beginning of the results section. (page 5, line 125-131)

“An overwhelming majority of HCPs and public health academics who made individual submissions (83%) endorsed the role of NVPs as a smoking cessation and/or harm reduction tool. In contrast, of the submissions from government and peak health bodies, 21 (91%) recommended against relaxing current restrictions until more evidence on efficacy and long term safety is available. Only one peak health body, the RANZCP, endorsed NVPs as a tool for harm reduction. There were commonalities among government agencies, peak health bodies and health charities in the way the problem was framed and understood, which we discussed in detail below.”

Third, the authors might want to comment on areas that might not have been covered well. For example, it seems likely there was acceptance of the success of existing regulation which has yielded slow but steady progress in reducing smoking rates, but with little emphasis given to the relatively slow progress that has been made and urgency to change the slow trajectory of change predicted. In other words, it appears there that the status quo is acceptable. Forth, there appears to have been no discussion of how accepting a smoking harm reduction might work synergistically with the well-established demand reducing policies already implemented in in Australia.

Response: We have now included one paragraph to address the comments given. We have discussed the unintended consequences of maintaining the status quo, which is endorsed as the appropriate policy response by the majority of policy actors. We have also discussed the complementarity of tobacco harm reduction with tobacco demand and supply reduction. (page 16, line 578-589)

 “Although the prevalence of daily smoking among adult Australians decreased from 24.3% in 1991 to 12.8% in 2013, the decline over the last few years has not been significant with only a reduction of 0.6% seen between 2013 and 2016 (from 12.8% in 2013 to 12.2% in 2016) and smoking remains highly prevalent among many priority populations, including people on low incomes, Indigenous peoples and people with mental illnesses. This is despite Australia being an international leader in implementing tobacco control strategies which are largely focused on reducing demand and supply of tobacco products. Although the National Tobacco Strategy 2012–2018 states that harm reduction is one of the three pillars of the strategy, harm reduction has not been widely implemented, especially in relation to lower risk substitutes for cigarettes. Although NVPs are not completely safe, evidence suggests that they are less harmful than combustible cigarettes (refs). Thus, controlled availability of NVPs could provide a harm reduction tool for unwilling or hard-to-quit smokers. Access to a lower risk substitute for cigarettes could complement demand and supply reduction strategies. However, Australia’s current regulatory framework, however, puts severe restrictions on access to NVPs, with potentially heavy penalties for possession and use without a prescription, while allowing nicotine in the form of smoked tobacco products to be sold as consumer products. One potential consequence of maintaining greater restrictions on nicotine in less harmful forms compared to cigarettes, is that some smokers will continue smoking rather than switching to the lower risk option. Hence, overly cautious regulation also carries risks such as potentially perpetuating smoking.”

Fifth, the debate about loosing regulations affecting access to nicotine vaping products in Australia appears not to consider the wishes of current smokers. It would be helpful to cite polling data reflective of consumer attitudes about regulating vaping in Australia.

Response: We have included a paragraph in the discussion and cited articles on consumer’s policy support regarding NVPs. (page 16 line 597-602)

“Moreover, the current regulatory framework does not seem to give due consideration to the views of consumers who are using or want to use NVPs as a cessation or harm reduction tool. A number of studies have reported that consumers prefer regulatory options that will not impede their ability to obtain and use NVPs.”

Finally, on a minor issue, I found the term "peak" health bodies unfamiliar. It would help the reader to have a definition of what the authors mean by peak health body since this term is frequently used in the manuscript.

Response: Peak health bodies are organisations representing health and medical professionals. We have included this in the introduction section (page 2, line 81-82)

Round 2

Reviewer 2 Report

The authors have done a nice job responding to reviewer comments.  No further changes are recommended.